# Area Entropy and Quantized Mass of Black Holes from Information Theory

**DOI:** 10.3390/e23070858

**Published:** 2021-07-03

**Authors:** Dongshan He, Qingyu Cai

**Affiliations:** 1College of Physics & Electronic Engineering, Xianyang Normal University, Xianyang 712000, China; hfrnsm@163.com; 2School of Information and Communication Engineering, Hainan University, Haikou 570228, China; 3State Key Laboratory of Magnetic Resonance and Atomic and Molecular Physics, Innovation Academy for Precision Measurement Science and Technology, Chinese Academy of Sciences, Wuhan 430071, China; 4Peng Huanwu Center for Fundamental Theory, Hefei 230026, China

**Keywords:** black hole entropy, information theory, Rayleigh criterion

## Abstract

In this paper, we present a derivation of the black hole area entropy with the relationship between entropy and information. The curved space of a black hole allows objects to be imaged in the same way as camera lenses. The maximal information that a black hole can gain is limited by both the Compton wavelength of the object and the diameter of the black hole. When an object falls into a black hole, its information disappears due to the no-hair theorem, and the entropy of the black hole increases correspondingly. The area entropy of a black hole can thus be obtained, which indicates that the Bekenstein–Hawking entropy is information entropy rather than thermodynamic entropy. The quantum corrections of black hole entropy are also obtained according to the limit of Compton wavelength of the captured particles, which makes the mass of a black hole naturally quantized. Our work provides an information-theoretic perspective for understanding the nature of black hole entropy.

## 1. Introduction

In the early 1970s, Bekenstein conjectured that black holes must have entropy or the second law of thermodynamics will be violated. The entropy of a black hole was suggested as a monotone increasing function of its area, since the area of a black hole never decreases in the classical theory [1]. The simplest monotone increasing function of the area is a proportional function, i.e., $S_{bh}=\gamma A$, where γ is a constant and *A* is the area of the black hole at its event horizon. When quantum field theory of curved spacetime was applied to black holes, Hawking discovered that black holes can emit particles [2], called Hawking radiation. The area entropy of black holes was then reaffirmed, and the coefficient γ can be easily determined, $S_{BH}=k_{B}A/4l_{p}^{2}$, where $k_{B}$ is the Boltzmann constant, $A=4\pi r_{M}^{2}$ is the area of the black hole, and $l_{p}=\sqrt{\hbar G/c^{3}}$ is the Planck length. There are different methods for deriving the entropy of a black hole [1,3,4,5,6,7], while the physical origin of black hole entropy is still a puzzle.

Entropy measures the amount of uncertainty of a physical system. Generally, more information implies less uncertainty [8], which is sometimes expressed simply in a quantitative statement: information is negative entropy. The more information gained, the less uncertainty or entropy a system has. Likewise, the entropy of black holes means information about a black hole is missing. Due to the no-hair theorem, all information about the matter in the hole is missing for an outside observer except the total mass, angular momentum, and charge of the black hole. After a particle with mass *m* falls into a black hole, the entropy of the black hole increases as $\Delta S=8\pi Mm+4\pi m^{2}$, and it depends not only on the particle *m*, but also on the black hole itself, which is quite similar to that of the resolution of an optical instrument. In this paper, the area entropy of a black hole is derived from the analogy between a black hole and a camera. The quantum corrections of black hole entropy can be naturally achieved with the upper bound of wavelength of captured particles, which gives an extra logarithmic term of the Bekenstein–Hawking entropy. Finally, the quantized mass spectrum of a black hole is presented.

## 2. Deriving Black Hole Area Entropy from the Rayleigh Criterion

In information theory, information gained from a physical system is precisely equal to the amount of the decrease of entropy before and after the measurement. The information gained, in general, is not automatically equal to the amount of information a system possesses. For example, when one wants to preserve the information of a scene, a picture may be taken with a camera. The maximum information captured by a camera, $I_{\mathrm{camera}}$, is limited by the resolution $R_{e}$ of the camera. The angular resolution (reciprocal of the minimum resolvable angle θ=1.220λ/D) describes the ability of the image-forming device, such as an optical telescope, a microscope, a camera, or an eye, to distinguish the small details of an object. For a single optical telescope, its resolution is determined by both the aperture of the telescope and the wavelength of the light source. Normally, the Rayleigh criterion is used to estimate the angular resolution of an optical imaging system, and the resolution (resolving power) of a telescope is approximated to
(1)$$R_{e}=\frac{D}{1.22\lambda }.$$
Here, *D* is the diameter of the telescope objective, and λ is the wavelength of the source light. It is obvious that the larger the resolution $R_{e}$ is, the greater the amount of information that can be obtained. The maximal information that an optical instrument can gain is proportional to its resolution, $\Delta I\propto R_{e}$.

The gravitational field of the massive object bends the light passing through its vicinity, which is called the gravitational lensing effect [9]. Gravitational lensing is an important probe of astronomical observations in cosmology [10]. Analogously, black holes can be considered as optical instruments, and the maximal information a black hole can gain from a particle (of mass *m*) outside is proportional to its resolution. In 1999, F. Scardiglidesigned a gedanken experiment to get an “image" of a black hole by looking at photons scattered by the black hole. Then the generalized uncertainty principle is obtained by using the Heisenberg principle and the resolution of optical instruments [11]. According to the no-hair theorem, the inner state of a black hole is not known to anyone, except for its macroscopic parameters, mass, charge, and angular momentum. After a particle *m* falls into a black hole *M*, the information about particle *m* will disappear. The information for the combined system composed of the particle and the black hole decreases, while their total entropy increases. Due to the relation between information and entropy, the amount of increased entropy of the black hole is
$$\Delta S_{bh}=-\Delta I_{m}.$$
The increase of the entropy for the black hole can hence be obtained as
(2)$$\Delta S_{bh}\propto \frac{D}{\lambda_{C}},$$
where $D=4GM/c^{2}$ is the diameter of the event horizon of the black hole. Since the wavelength of the particle is limited by its Compton wavelength $\lambda_{C}=h/mc$, the increase of the black hole entropy can thus be rewritten as
(3)$$\Delta S_{bh}\propto \frac{2G}{\pi c\hbar }Mm.$$

There are lots of massless photons falling into the black hole, which will increase the mass as well as the entropy of the black hole. For a photon, the relationship between its wavelength λ and its energy ε is λ=2πℏc/ε. When a photon with energy ε enters the black hole, the increment of the black hole entropy is
(4)$$\Delta S_{bh}\propto \frac{2G}{\pi c^{3}\hbar }M\varepsilon .$$
According to the Einstein relationship between mass and energy, the increased mass of the black hole is $dM=\varepsilon /c^{2}$.

Therefore, no matter what kinds of particles falling into a black hole, by integrating Equations (Equation 3) or (Equation 4), one obtains
$$\begin{array}{cc} S_{bh} & \propto \int_{0}^{M}\frac{2G}{\pi c\hbar }MdM,\\ \; & \propto \frac{A}{16\pi^{2}l_{p}^{2}}, \end{array}$$
where *A* is the area of the black hole. This result is consistent with the area law of black hole entropy. In more detail, a prefactor of order unity is found to be missing in the equation above based on rough estimates. This missing factor can be determined by a more elaborate approach, such as that already found by Hawking with the discovery of the famous Hawking radiation [2]. Comparing with the Bekenstein–Hawking entropy, $S_{BH}=k_{B}A/4l_{p}^{2}$, the missing prefactor on the right side of Equations (Equation 2) and (Equation 3) should be $4\pi^{2}k_{B}$. For simplicity, the Planck unit system is adopted later.

## 3. Quantum Corrections of Black Hole Entropy

For an optical imaging system, if the wavelength λ of the light is longer than the diameter of the aperture *D*, no information can be obtained by the optical system due to diffraction. Similarly, a black hole may not capture information when the Compton wavelength of the falling particle is longer than the diameter of the black hole. So, we can reasonably assume that the maximum Compton wavelength of particles falling into a black hole is limited by the diameter of the black hole $\lambda_{max}∼D$. The maximum Compton wavelength of falling particles indicates that there is a minimum of information that a black hole can capture, ${\left(\Delta S_{bh}\right)}_{min}$. Suppose that
(5)$${\left(\Delta S_{bh}\right)}_{min}=\beta .$$
According to the Rayleigh criterion in Equation (Equation 3), the minimum mass of a particle that falls into a black hole is
(6)$${\Delta M}_{min}=\frac{\beta }{8\pi M}.$$
When supposing that the frequency of a Hawking radiation is proportional to the temperature of the black hole, one can get a similar result [12].

After N−1 minimum particles have fallen into a black hole (it should be noted that the mass of the N−1 particles are different; see Table 1), the entropy of the black hole becomes
(7)$$S_{bh}\left(N\right)=S_{bh}\left(1\right)+\left(N-1\right)\beta ,$$
where the entropy and the mass of the initial black hole are $S_{bh}\left(1\right)$ and M(1), respectively. According to Equation (Equation 6), one can obtain
(8)$$\sum_{n=1}^{N}8\pi M\left(n\right)\Delta M\left(n\right)=N\beta .$$
Here, M(n) represents the mass of the black hole when the *n*-th particle falls into the black hole, and ΔM(n)=M(n+1)−M(n). Inserting Equation (Equation 8) into Equation (Equation 7), we obtain
(9)$$S_{bh}\left(N\right)=\sum_{n=1}^{N}8\pi M\left(n\right)\Delta M\left(n\right)+S\left(1\right)-\beta .$$

To investigate the relationship between black hole entropy and the mass, one needs to calculate the summation in Equation (Equation 9). This gives two identities for series M(n) [13],
(10)$$\begin{array}{cc} \sum_{n=1}^{N}M\left(n\right)\Delta M\left(n\right) & =\sum_{n=1}^{N}M\left(n\right)M\left(n+1\right)-\sum_{n=1}^{N}M^{2}\left(n\right), \end{array}$$
(11)$$\begin{array}{cc} \sum_{n=1}^{N}M\left(n+1\right)\Delta M\left(n\right) & =\sum_{n=1}^{N}M^{2}\left(n+1\right)-\sum_{n=1}^{N}M\left(n\right)M\left(n+1\right). \end{array}$$
Using the two identities above, we can get
(12)$$\sum_{n=1}^{N}M\left(n\right)\Delta M\left(n\right)=M^{2}\left(N+1\right)-M^{2}\left(1\right)-\sum_{n=1}^{N}M\left(n+1\right)\Delta M\left(n\right).$$
Inserting M(n+1)=M(n)+β/8πM(n) into Equation (Equation 12), we obtain
(13)$$\sum_{n=1}^{N}M\left(n\right)\Delta M\left(n\right)=\frac{M^{2}\left(N+1\right)-M^{2}\left(1\right)}{2}-\frac{\beta }{16\pi }\sum_{n=1}^{N}\frac{\Delta M(n)}{M(n)}.$$
Next, inserting Equation (Equation 13) into Equation (Equation 9), we have that
(14)$$S_{bh}\left(M\right)=4\pi \left[M^{2}\left(N+1\right)-M^{2}\left(1\right)\right]-\frac{\beta }{2}\sum_{n=1}^{N}\frac{\Delta M(n)}{M(n)}+S\left(1\right)-\beta .$$

Up to this point, our calculations have been rigorous and no approximations were taken. In Equation (Equation 14), the leading order of black hole entropy is $S=4\pi M^{2}$, but it is difficult to give a simple and accurate result for the second term on the right hand of Equation (Equation 14). Thus, we carry out an approximate calculation for the last summation of Equation (Equation 14) for M≫1.
(15)$$S_{bh}\left(M\right)≃4\pi M^{2}-\frac{\beta }{4}\ln M^{2}+O\left(M^{-1}\right).$$
Here, we have set M(1)=1, and $S_{bh}\left(1\right)=4\pi$. It is not very surprising that there is a logarithmic correction of a Schwarzschild black hole in Equation (Equation 15), which is consistent with the results from string theory or loop quantum gravity [14,15,16] and the results from Bose statistics [17]. The coefficients of the logarithmic correction term are different in different quantum gravity theories. In our calculations, the coefficient depends on the minimum unit of entropy of black holes.

## 4. Quantized Entropy and Mass of Black Holes

According to Equation (Equation 7), the quantized entropy of a black hole can be obtained when taking β=4π,
(16)S(N)=4πN.
In this case, black hole entropy can be taken only as an integer multiple of 4π, (Taking ΔM=1/4M, one can get β=2π, and S(N)=2πN [18].) not as a continuous value. The mass of the black hole cannot take an arbitrary value either, because the minimum mass of a particle falling into the black hole is limited. With the recurrence relations M(n+1)=M(n)+1/2M(n) from Equation (Equation 6), the quantized mass of a black hole can be expressed as
(17)$$M=M\left(1\right)+\frac{1}{2M(1)}+\frac{1}{2\left(M\left(1\right)+\frac{1}{2M(1)}\right)}+\cdots .$$

In Table 1, we list the first five terms of the quantized entropy and mass of a black hole. It is difficult to obtain a rigorous analytical expression for the quantized mass of a black hole. For N≫1, we find that the quantized mass of black holes is approximately equal to
(18)$$M\left(N\right)≃\sqrt{N+\frac{\ln N}{4N}}≃\sqrt{N},$$
which is coincident with the results in [19,20,21] and the result from the quantum description of a black hole [22]. When *N* is very large, we have $\Delta M=1/2M\left(N\right)≃1/2\sqrt{N}\to 0$. This indicates that for a large black hole, its mass spectrum is approximately continuous. The black hole can either absorb a small particle and jump to a nearby quantum state, or absorb a large particle and jump to a distant quantum state. It is interesting that $S_{bh}\left(N\right)=4\pi N\neq A/4$ for a very small black hole, which shows that the area law of black hole entropy may be an approximate law for a massive black hole but the quantum corrections cannot be ignored for a small black hole.

Numerical solutions of the quantized mass for a black hole are shown in Figure 1. The exact solution and the approximate solution of the quantized mass of a black hole are almost identical. In Figure 2, numerical solutions show that the logarithmic correction entropy is closer to the quantized entropy than to the Bekenstein–Hawking entropy $S_{BH}$. It should be noticed that the logarithmic correction entropy is not a monotonic increasing function of mass. For a very small black hole($M<\sqrt{\beta /16\pi }$), one finds that $dS_{bh}/dM<0$, which seems to violate the second law of black hole thermodynamics. In fact, Equation (Equation 15) only holds when *M* is far greater than 1, so it does not really conflict with the second law of black hole thermodynamics.

## 5. Discussion and Conclusions

We have shown that the area law of black hole entropy can be obtained when comparing the black hole to an optical camera, which implies that the black hole entropy is information entropy rather than thermodynamic entropy. The quantum corrections to the Bekenstein–Hawking entropy were rediscovered for M≫1, and we found that the coefficient of the correction term strongly depends on the minimum unit of black hole entropy. The quantized entropy will transition to the Bekenstein–Hawking entropy when the minimum entropy unit of a black hole tends to zero (β→0).

It is interesting that the Planck constant *ℏ* appears in the denominator of Bekenstein–Hawking entropy. As the Planck constant approaches zero, the Bekenstein–Hawking entropy diverges. This definitely means that the area entropy of a black hole has no corresponding quantity in classical theory. Therefore, a complete quantum gravity theory is desired to study the properties of black holes entirely. Since the area entropy and quantized mass of black holes can be derived from the information theory, it can be expected that information theory may play an important role in establishing a complete quantum gravity theory.

## Figures and Tables

**Figure 1 entropy-23-00858-f001:**
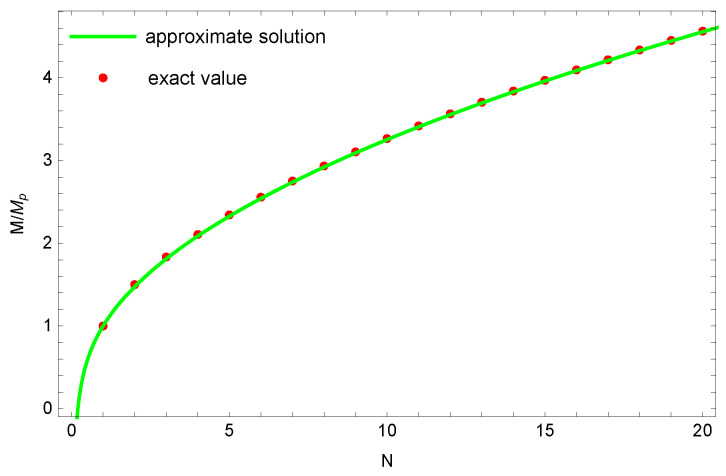
The quantized mass of black holes. The green solid line and red dots represent the approximate solution and exact solution of quantized mass for a black hole, respectively. The approximate solution is based on Equation (Equation 18), and the exact mass is based on the summation of Equation (Equation 17). $M\left(1\right)=M_{P}$, and β=4π.

**Figure 2 entropy-23-00858-f002:**
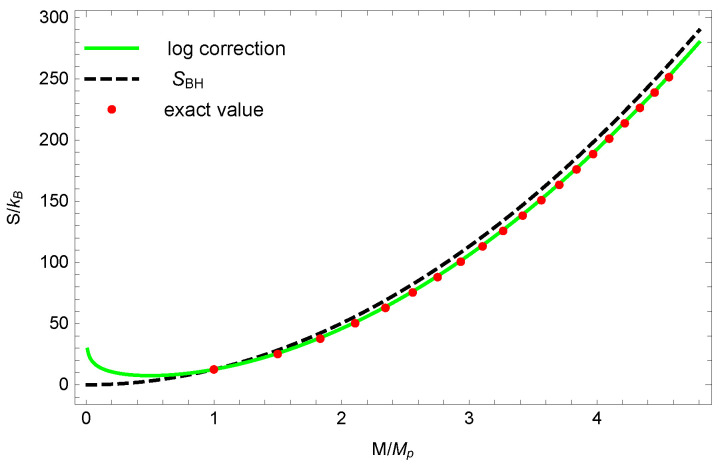
Black hole entropy with different approximations. The black dashed line, green solid line and red dot represent the Bekenstein–Hawking entropy, the logarithmic correction entropy and the quantized entropy respectively, with $M\left(1\right)=M_{P}$ and β=4π.

**Table 1 entropy-23-00858-t001:** The quantized entropy and mass of a black hole. Here $M\left(1\right)=M_{P}$ and β=4π.

*N*	1	2	3	4	5	⋯
S(N)	4π	8π	12π	16π	20π	⋯
M(N)	1	$\frac{3}{2}$	$\frac{11}{6}$	$\frac{139}{66}$	$\frac{21,499}{9174}$	⋯

## Data Availability

The data presented in this study are available through contacting the corresponding author: qycai@wipm.ac.cn.

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
