# Peer review of "Area Entropy and Quantized Mass of Black Holes from Information Theory"

_entropy, 2021, doi:10.3390/e23070858_

Round 1

Reviewer 1 Report

The main claim is that the analysis from the paper takes into account only massive particles falling into black hole (BH). In opposite there are a lot of photons with zeroth mass reach the BH. They increase the energy of BH therefore its mass. In the book of Frolov & Novikov "Physics of black holes" one can find the direct formula with percentage of each type. Therefore the ideas from the paper are only rough estimations. I think that the comment of how these photons are taken into account is necessary.

Author Response

Dear reviewer,

Thank you for your valuable advice. Massless photons falling into a black hole do increase the mass and entropy of the black hole. For a photon falls into the black hole, the resolution of the black hole determined by the black hole’s diameter and the wavelength of the photon. Using the same way as massive particle, we can get the entropy that increased by photons. Interestingly, the result is the same as before. In the revised draft, we added the influence of photon on the entropy of black hole, please see the attachment.

Yours sincerely,

Dongshan He, Qing-yu Cai

Reviewer 2 Report

The manuscript derives the Bekenstein-Hawking entropy by considering the black hole as an imaging device which hides the information about the particles it absorbes. Quite interestingly, the rather ubiquitous logarithmic correction is also recovered from the same argument. I find the approach inspiring and I believe the paper deserves to be published.

I would only suggest the authors to consider connections with other works in the literature. In details:
1) the analogy between a black hole and a lens closely resembles the way the Generalised Uncertainty Principle (GUP) is introduced, e.g. in Scardigli, Phys. Lett. B 452 (1999) 44 [hep-th/9904025] (the literature about the GUP is rather vast and similar arguments could be found in even older papers cited therein);
2) the area quantisation N ~ M^2 is obtained in a similar fashion in Dvali, Gomez, Mukhanov, arXiv:1106.5894, and is at the heart of the corpuscular picture (see, e.g. Dvali and Gomez, Fortsch. Phys. 61 (2013) 742 [arXiv:1112.3359]);
3) the corpuscular description of black holes also reproduces a logarithmic correction, as shown in Casadio et al, Phys. Rev. D 91 (2015) 124069 [arXiv:1504.05356]. Finally
3) a recent derivation of N ~ M^2 based on quantum gravity is also found in Casadio, arXiv:2103.14582.

The above list is of course not exhaustive and I would leave it to the authors whether to comment about those particular approaches.

Author Response

Dear reviewer,

Thank you for your valuable advice. In our previous work, we haven't seen the literature you mentioned. Some of this work is really interesting and important, e. g. in Scardigli, Phys. Lett. B 452 (1999) 44 , the generalized uncertainty principle was obtained by using the Heisenberg principle and the resolution of optical instruments, which is a little bit like what we thought. We have supplemented these important references in the revised draft, please see the attachment.

Yours sincerely,

Dongshan He, Qing-yu Cai

Round 2

Reviewer 1 Report

I think that the additions help to understand the basic idea, the paper could be accepted for publication.